# Chemerin as a Driver of Cardiovascular Diseases: New Perspectives and Future Directions

**DOI:** 10.3390/biomedicines13061481

**Published:** 2025-06-16

**Authors:** Anna M. Imiela, Jan Stępnicki, Patrycja Sandra Zawadzka, Angelika Bursa, Piotr Pruszczyk

**Affiliations:** Department of Internal Diseases and Cardiology, Centre for Management of Venous Thromboembolic Disease, Medical University of Warsaw, 02-005 Warsaw, Polandpiotr.pruszczyk@wum.edu.pl (P.P.)

**Keywords:** adipokine, cardiovascular diseases, chemerin, hypertension, obesity

## Abstract

In recent years, the immune system has emerged as a key player in the development of atherosclerosis, heart failure, venous thromboembolism, and systemic hypertension. Obesity and related cardiovascular diseases (CVDs) remain the leading global cause of death. Adipokines—hormones produced by adipose tissue—exert diverse endocrine and immunomodulatory effects. Among them, chemerin, discovered in the early 20th century, is a chemotactic molecule that recruits dendritic cells, endothelial cells, macrophages, and lymphocytes during early immune responses. It regulates cell migration and vascular homeostasis. Dysregulated adipokine profiles contribute to chronic inflammation, insulin resistance, metabolic syndrome, and impaired blood pressure control. This review explores chemerin’s potential role in CVD pathogenesis, focusing on its immunomodulatory functions, impact on vascular inflammation, and endothelial dysfunction. The presented work also examines recent findings on chemerin’s diagnostic and therapeutic potential in cardiovascular health.

## 1. Introduction

In the past several years, it has become evident that the immune system plays a pivotal role in the pathogenesis of atherosclerosis, heart failure, venous thromboembolism, and even systemic hypertension [1,2,3]. Obesity and concomitant cardiovascular diseases (CVD) are the most common cause of death worldwide. Adipokines, which are produced in the adipose tissue, possess a wide plethora of endocrine and immunomodulatory functions [4]. Discovered at the beginning of the 20th century, chemerin is a chemotactic molecule that is involved in the early recruitment of dendritic cells (DCs), endothelial cells (ECs), macrophages, and lymphocytes [5]. Chemerin possesses multiple functions regulating cell migration and vasculature homeostasis [6]. Dysregulation of the adipokines profiles results in a series of pathological changes such as low-grade inflammation, insulin resistance, metabolic syndrome, and poor blood pressure control.

The aim of this manuscript is to address the potential detrimental role of chemerin in the pathogenesis of cardiovascular disorders. This work concentrates on the immunomodulatory effects of chemerin and its implications in vasculature, pro-inflammatory state development, and endothelium dysfunction. It places a central and explicit focus on chemerin’s immunomodulatory functions in CVD pathogenesis. Particular attention is given to chemerin’s function as a chemotactic agent essential for the initial recruitment of immune cells such as dendritic cells, macrophages, and lymphocytes. According to the latest research findings, this review aims not only to clarify the multifactorial role of chemerin in CVD pathophysiology but also its potential therapeutic and diagnostic applications. It advances the current understanding of chemerin’s involvement in CVDs by presenting updated perspectives and identifying key avenues for future research. Consequently, this work offers an updated and comprehensive synthesis that, through its specialized emphasis on immunomodulation, provides an outlook on chemerin’s role across various CVDs.

## 2. Methods

A comprehensive PubMed search was performed to analyze the articles published between 2007 and 2025 using the keywords “chemerin” and “cardiovascular disease”. Studies were selected with particular emphasis on recently published research. Relevant articles were thematically categorized into five major areas: inflammation, obesity, metabolic syndrome, hypertension, and atherosclerosis. Within each thematic category, studies were further divided into subgroups based on whether they involved animal models or human subjects. Findings were then compared to identify consistent conclusions and notable discrepancies, with a particular focus on their potential clinical relevance.

### Chemerin—Molecular Effect and Site of Action

Chemerin is a 16-kDa protein, an adipokine that was first discovered in white adipose tissue (WAT) in 2007. It is expressed mainly in the adipose tissue (AT) but also in the myocardium, lung tissue, and spleen [3,7,8]. It plays a key role in regulating vascular function, lipid metabolism, and inflammation, primarily through a calcium-dependent pathway [9]. Chemerin is encoded by the retinoic acid receptor responder protein 2 (*RARRES2*) gene and is initially secreted as pre-pro-chemerin. After the cleavage of a 20-amino-acid peptide, it is converted into pro-chemerin, which circulates in plasma [3,10]. The active form of chemerin is generated through further C-terminal cleavage by angiotensin-converting enzyme type 2 (ACEI-2) [11]. Chemerin primarily exerts its effects through chemerin-like receptor 1 (CMKLR1), also known as chemokine receptor-like 1, which is expressed in immune cells such as dendritic cells (DCs), endothelial cells (ECs), macrophages, and AT [3,12]. Upon binding to CMKLR1, chemerin activates the Gi protein, reducing cyclic adenosine monophosphate (cAMP) levels and inducing phosphorylation of nuclear factor kappa B (NF-κB) and extracellular signal-regulated kinase 1/2 (ERK1/2) [13].

A secondary receptor, G protein-coupled receptor 1 (GPR1), is predominantly expressed in the ovaries, placenta, testicles, and brain [13,14]. Although GPR1 shares over 40% sequence homology with CMKLR1, its biological effects are significantly weaker [13,14]. Additionally, the CCRL2 receptor modulates chemerin activity by increasing its local concentration and enhancing CMKLR1 signaling, which may play a role in physiological and inflammatory processes (Figure 1) [15]. At the cellular level, chemerin promotes cell cycle progression. Knockdown of either chemerin or CMKLR1 significantly reduces cyclin A2 and cyclin B2 levels, leading to an arrest in adipogenesis at the M/G2 phase [16]. As adipocytes mature, chemerin secretion increases, further stimulating proliferation and creating a positive feedback loop [17].

Figure 1 summarizes the most relevant chemerin mechanism of action; the main source of chemerin synthesis is adipose tissue and the liver.

## 3. The Physiological Role of Chemerin in Adipocytes

AT produces pro-inflammatory proteins that regulate body homeostasis. However, WAT disrupts adipokine function, leading to severe complications [18]. Under physiological conditions, chemerin regulates glycemic balance and adipocyte development. However, obesity disrupts chemerin secretion and function, contributing to metabolic imbalance [19]. Fat tissue comprises various cell types, including blood, immune, and EC, but adipocytes are the most abundant [20]. Adipogenic differentiation is regulated by numerous proteins, including hormones, retinoids, and adipokines [21,22,23].

### 3.1. Chemerin in Glucose Metabolism

Adipocyte metabolism is closely linked to plasma insulin levels [24]. In pancreatic β-cells, insulin undergoes expression and post-translational modifications [25]. Both chemerin and its receptor, CMKLR1, are localized in β-cells and play a crucial role in glucose homeostasis [26]. Increased insulin levels promote the translocation of insulin-regulated glucose transporter 4 (GLUT4) receptors to the adipocyte membrane, stimulating lipogenesis [27]. Chemerin enhances insulin-stimulated glucose uptake by increasing insulin sensitivity, thereby contributing to fatty acid and triglyceride synthesis [28]. Chemerin-deficient mice were characterized by impaired glucose-dependent insulin secretion (GSIS) in chemerin-deficient mice [26]. Notably, *chemerin* or *CMKLR1* knockdown in preadipocytes was associated with disruption of adipocyte differentiation and a significant reduction in gene expression involved in glucose and lipid metabolism [17]. Furthermore, CMKLR1 deficiency diminishes adipocyte glucose uptake and reduces body fat, highlighting chemerin’s crucial role in adipose tissue homeostasis [29,30]. However, studies have yielded conflicting results. Data coming from the mouse fat cell line-3T3-L1 adipocytes have demonstrated that chemerin presents a potential ability to both enhance and reduce insulin-stimulated glucose uptake, indicating inconsistent results [31]. Chemerin decreases insulin-stimulated glucose uptake, especially in the skeletal muscle cells [32]. In other mouse models, acute treatment with exogenous chemerin has worsened glucose tolerance due to the lower serum insulin levels and decreased glucose uptake in adipose tissue and the liver in in obese/diabetic but not normoglycemic mice [33].

### 3.2. Chemerin in Lipid Metabolism

Chemerin is known to regulate both adipogenesis (formation of fat cells) and lipolysis (breakdown of fat). In the bovine intramuscular adipocytes, chemerin induces triglyceride hydrolysis, leading to glycerol and free fatty acid efflux [30]. Additionally, chemerin upregulates leptin, a lipolytic adipokine, further confirming its role in fat breakdown [17,23]. However, inhibiting the ERK1/2 signaling pathway significantly reduces chemerin’s ability to promote adipose tissue breakdown, indicating that its lipolytic function depends on ERK1/2 activation [34]. The ERK1/2 pathway upregulates lipolysis genes and suppresses adipogenesis markers.

### 3.3. Chemerin in Angiogenesis and Vascular Function

Adipose tissue (AT) is a highly vascularized structure with dense blood vessels net playing a crucial role in supporting adipocyte function, growth, and survival. Vascular regulation helps maintain the local AT environment, responding to factors like hypoxia and acidosis. Simultaneous signals from adipocytes promote angiogenesis in AT. In physiological conditions, ECs possess abilities to dynamically shift into a proliferative state in response to angiogenic or metabolic signals [35]. In addition, pro-angiogenic factors, such as lepton, VEGF, and adiponectin, are produced by AT and act in the self-autocrine loop modulating vascular cell growth [35]. Interestingly, precursor cells capable of becoming ECs or adipocytes have been identified within AT vasculature, highlighting a close developmental and functional link between adipocytes and ECs. Additionally, fat tissue regulation relies on angiogenesis, as blood vessel formation supports adipose cell maturation by delivering nutrients and fulfilling metabolic demands [35]. Angiogenesis is driven by EC proliferation and migration [36]. The receptor CMKLR1, expressed on ECs, activates key neovascularization pathways—the phosphatidylinositol 3-kinase/protein kinase B (PI3K/Akt), which promotes EC migration and mitogen-activated protein kinase (MAPK), which regulates cell proliferation [37,38,39]. Chemerin also activates matrix metalloproteinases, specifically gelatinase A (MMP-2) and gelatinase B (MMP-9), which help organize ECs into tubular structures [37,40]. Thus, chemerin-driven angiogenesis plays a crucial role in adipocyte formation and expansion [41].

## 4. Chemerin in Obesity-Induced Inflammation

Inflammation is increasingly recognized as a central factor in the development and progression of CVDs, largely through the activation of the body’s innate and adaptive immune responses. Persistent inflammatory activity can damage ECs and facilitate the formation of atherosclerotic plaques, leading to adverse cardiac remodeling and fibrosis. Additionally, a growing body of research has established a strong link between inflammation and the pathophysiology of obesity. These processes are driven by cytokine production and the recruitment of inflammatory cells via chemokines [42]. Chemerin, a chemokine with chemotactic properties, binds to CMKLR1 to regulate immune responses [43,44]. CMKLR1 is highly expressed on antigen-presenting cells (APCs), including macrophages and immature DCs, and also plays a role in natural killer (NK) cell activity [45]. Elevated serum chemerin levels have been observed in various inflammatory diseases, such as non-alcoholic fatty liver disease (NAFLD), rheumatoid arthritis, and bowel inflammatory conditions [46,47,48]. Notably, obesity is strongly associated with increased circulating chemerin levels [49,50].

Studies have shown that macrophage, lymphocyte, and neutrophil accumulation in adipose tissue is a hallmark of obesity, with immune cell infiltration correlating positively with excess fat [51,52,53]. High-fat deposition promotes the accumulation of pro-inflammatory M1 macrophages while reducing anti-inflammatory M2 macrophages, leading to systemic inflammation [54]. Chemerin acts as a chemoattractant for M1 macrophages, linking obesity to chemerin-driven inflammation [55,56]. Macrophages secrete key inflammatory markers—tumor necrosis factor-α (TNF-α), interleukin 1β (IL-1β), and interleukin 6 (IL-6)—which are elevated in WAT expansion [55,57,58].

Interestingly, the inflammatory state is amplified by excessive chemerin production by TNF-in AT [8]. This highlights chemerin’s crucial role in obesity-induced inflammatory cascades. Serum chemerin levels positively correlate with other pro-inflammatory adipokines, including leptin and resistin [59]. Additionally, anti-inflammatory adiponectin is downregulated in association with elevated circulating chemerin levels [60]. Chemerin is speculated to regulate the synthesis of inflammatory mediators in obese patients, such as monocyte chemoattractant protein-1 (MCP-1), plasminogen activator inhibitor-1 (PAI-1), and von Willebrand factor (vWF) [61]. In response to elevated IL-6 levels, the liver triggers the secretion of high-sensitivity C-reactive protein (hs-CRP), a key marker of inflammation in obesity [62]. IL-6 synthesis is also upregulated in adipocytes during inflammation, contributing to chronic, low-grade inflammation linked to fat mass [62,63]. Moreover, hs-CRP levels rise with increasing chemerin concentrations, reinforcing chemerin’s role in promoting inflammation [64]. Inflammation, predominantly induced in WAT, may contribute to chronic systemic inflammation affecting multiple organs [65].

Traditionally, obesity is strongly related to hypertension, dyslipidemia, and type 2 diabetes mellitus (T2DM) [66,67,68,69]. Interestingly, glucagon-like peptide-1 (GLP-1) agonists, which suppress appetite and promote fat loss, can downregulate chemerin expression [19,70,71]. GLP-1 also exhibits notable anti-inflammatory properties, as research indicates it reduces inflammatory cytokine and marker levels [72,73]. Hypoxia, commonly observed in obesity, leads to adipocyte apoptosis [74,75,76]. M1 macrophages accumulate around dying fat cells, forming crown-like structures that exacerbate inflammation [77,78]. GLP-1 agonists mitigate inflammation by alleviating hypoxia [79]. In obese mice, treatment with a recombinant adenovirus engineered to induce high GLP-1 levels significantly reduced M1 macrophage expression in adipocytes. GLP-1 promoted fat oxidation and suppressed inflammatory responses [80]. Additionally, chronic inflammation and elevated chemerin levels in obesity may contribute to reduced GLP-1 secretion, potentially through TNF-α-induced suppression [81,82]. 

## 5. Chemerin and Obesity—Studies

### 5.1. Chemerin and Obesity—Data from Animals Models and Cells

While WAT is the primary site of chemerin secretion, this adipokine is also present in multiple tissues, including the liver, lungs, ovaries, and pituitary gland [5,49]. To investigate whether chemerin modulates adipocyte development and function, studies have been conducted in experimental mouse models and in vitro human cells [16,17]. During adipocyte differentiation, *chemerin* and *CMKLR1* expression increases. By activating the ERK1/2 pathway in adipocytes, chemerin binds and stimulates CMKLR1 via autocrine/paracrine signaling, promoting adipogenesis. Additionally, the knockout of *chemerin* or *CMKLR1* in the mouse’s preadipocytes prevented their development [16]. Gene deletion suppressed differentiation at the G2/M phase by reducing cyclin A2 and B2 levels, which are crucial for mitosis [83]. Experiments on human cells yielded similar results, indicating the existence of common pathways and mechanisms for adipogenesis that involve chemerin/CMKLR1 [16,17,83]. The study conducted by Ernst MC et al. has shown that *CMKLR1^−/−^* mice under both low-fat and high-fat diets have shown nearly 25% less food intake, lower body mass, and reduced fat mass compared to wild-type (WT) mice [29]. Additionally, *CMKLR1^−/−^* mice were glucose intolerant due to reduced insulin secretion in response to glucose, especially in the skeletal muscle and white adipose tissue. This suggests that chemerin and CMKLR1 are essential for adipogenesis and the increased fat mass associated with obesity [29].

Angiogenesis is required for both new fat cell formation (hyperplasia) and volume increase (hypertrophy). While hypertrophy is the primary driver of adipose tissue expansion, hyperplasia may demonstrate a secondary role. Neovascularization supports these processes by connecting adipose tissue with other organs, supplying nutrients, and removing metabolic waste [84,85]. Chemerin modulates endothelial cell migration, proliferation, and angiogenesis via activation of PI3K/Akt, ERK1/2, and MAPK pathways, leading to vascular remodeling and increased arterial stiffness [35]. Its pro-angiogenic potential has been confirmed in multiple in vivo and ex vivo experimental models, including the Matrigel plug assay, mouse corneal angiogenesis model, and rat aortic ring assay [86]. In the Matrigel plug assay, plugs containing chemerin, VEGF, or control solution were implanted subcutaneously in mice, followed by histological evaluation after two weeks [86]. Chemerin-exposed plugs exhibited significantly higher endothelial cell density compared to controls. Studies on C57BL/6 mice (WT mice) and Sprague Dawley rats (SD rats) have demonstrated chemerin’s ability to positively influence vessel growth. Specifically, chemerin treatment significantly increased neovascularization in the cornea of mice (49 ± 4 capillary number/cornea) compared to the control group (no capillaries). Similarly, rats showed a marked increase in vessel development in aortic rings (1220 ± 950 pixel/area) in comparison with untreated controls showing no vessel growth. To determine how chemerin promotes angiogenesis, studies have been performed on human cells [86]. Nakamura N. et al. have demonstrated that chemerin drives angiogenesis in human umbilical vein endothelial cells (HUVECs) [86]. Chemerin secretion has promoted the formation and proliferation of capillary-like structures as well as acted as a chemoattractant in migration experiments. Additionally, chemerin triggered the activation (phosphorylation) of Akt and ERK signaling pathways in HUVECs. Blocking the CMKLR1 receptor using small interfering RNA (siRNA) entirely suppressed chemerin’s ability to promote both cell migration and angiogenesis, while silencing chemokine receptor-like 2 (CCRL2) had no such effect. This suggests that chemerin’s pro-angiogenic and migratory effects on HUVECs are primarily mediated through CMKLR1 [86].

### 5.2. Chemerin and Obesity—Data from Humans

There are many tools to evaluate the amount of AT such as body mass index (BMI), waist-to-hip ratio (WHR), and waist circumference [87]. There is a significant correlation between higher antropometric and chemerin concentrations, suggesting a role for chemerin in obesity and adipose tissue distribution [88,89]. Subjects with a BMI over 30 kg/m^2^ have presented a considerably higher chemerin level (averaging 296.5 ng/mL) compared to those with a BMI under 25 kg/m^2^ (mean: 222.7 ng/mL) [49]. Furthermore, patients who have undergone bariatric surgery exhibit decreased circulating chemerin levels, reinforcing the strong correlation between chemerin concentration and fat tissue volume [32,90,91].

Given chemerin’s potential role in angiogenesis, and the fact that angiogenesis is vital for adipose tissue expansion, chemerin circulating levels may reflect its role in supporting the vasculature necessary for tissue growth. Treatment with human chemerin concentrations of 0.3 ng/mL increased total tubule length by 34% (*p* < 0.05), while concentrations of 1 and 3 ng/mL led to a 38% increase (*p* < 0.001) compared to control samples [41]. Studies on human umbilical cord vein cells (HUVECs) and human microvascular endothelial cells (HMECs) have demonstrated that chemerin positively influences vessel growth by stimulating endothelial tube formation and EC migration. Furthermore, chemerin enhances the synthesis of fatty acid-binding protein 4 (FABP4) and vascular endothelial growth factor (VEGF), further promoting adipogenesis and increasing adipose tissue vascular density [37,86,92].

Initially discovered in psoriatic skin lesions and inflamed human fluids, chemerin plays a key role in the infiltration and recruitment of DCs and macrophages [5,93,94]. WAT consists not only of adipocytes but also various immune cells, including macrophages, DCs, and lymphocytes (B-cells, T-cells, and natural NK cells) [95]. Research supports the role of chemerin as a chemoattractant for these immune cells, linking obesity with inflammation. CMKLR1 expression has been identified in the naïve and M1 macrophages, DCs, and NK cells [55,96,97].

Patients characterized with a higher BMI also exhibit elevated levels of chemerin, pro-inflammatory adipokine-leptin, and inflammatory markers such as TNF-α, hs-CRP, and IL-6 [59]. A positive correlation was observed between hs-CRP and chemerin concentration in the experimental models of non-diabetic obese (NDO) and non-diabetic non-obese patients. These markers were significantly higher in the NDO group compared to controls, further supporting chemerin’s pro-inflammatory role in obesity [60,98].

Generally, in mouse models, chemerin promotes adipocyte differentiation via CMKLR1, with both increasing during adipogenesis through ERK1/2 signaling. Deletion of chemerin or CMKLR1 halts differentiation at the G2/M phase by reducing cyclins A2 and B2. CMKLR1-deficient mice show lower body mass, food intake, and fat accumulation under various diets but develop glucose intolerance due to impaired insulin secretion, highlighting chemerin’s metabolic role. Chemerin also enhances angiogenesis in vivo, as shown in Matrigel plug, aortic ring, and corneal assays, by activating PI3K/Akt, ERK1/2, and MAPK pathways—effects lost when CMKLR1 is silenced, confirming its key role.

Human studies link elevated serum chemerin to a higher BMI, waist-to-hip ratio, and adiposity. Obese individuals (BMI > 30) have significantly higher chemerin than lean individuals, with levels decreasing post-bariatric surgery. In vitro, chemerin stimulates endothelial migration, tube formation, and angiogenic marker expression (VEGF, FABP4) in HUVECs and HMECs via Akt and ERK pathways—effects abolished by CMKLR1 silencing. Chemerin also contributes to obesity-related inflammation by recruiting macrophages, dendritic cells, and NK cells and correlates with pro-inflammatory markers such as TNF-α, IL-6, and hs-CRP.

## 6. Chemerin and the Metabolic Syndrome

Metabolic syndrome (MeS) is a severe clinical condition that increases the risk of severe health problems, including T2DM and CVDs. Key risk factors include elevated blood glucose, hypertension, abdominal obesity, hypertriglyceridemia, and low high-density lipoprotein (HDL) levels [99]. In addition to genetic predisposition, lifestyle factors also influence MeS development [100,101].

Notably, high visceral adipose tissue mass contributes to MeS pathogenesis by triggering metabolic disturbances and a pro-inflammatory state [102]. This pro-inflammatory state disrupts adipokine expression in fat tissue, exacerbating metabolic dysfunction [103]. Chemerin is a critical regulator of glucose and lipid homeostasis, and its dysregulation can disrupt metabolic balance [19]. Several studies have highlighted a strong correlation between elevated chemerin levels and MeS, further supporting its role in metabolic dysfunction [49,50,104,105].

Insulin resistance occurs when tissues become unresponsive to normal insulin levels [106]. The Homeostatic Model Assessment for Insulin Resistance (HOMA-IR) is commonly used to assess insulin resistance [107,108]. High HOMA-IR, along with increased circulating chemerin levels, are characteristic of patients with MeS [105,109]. Numerous studies have reported a positive correlation between serum chemerin levels and insulin resistance [110,111,112]. One of the primary causes of reduced insulin sensitivity is inflammation, driven by obesity-induced macrophage infiltration in WAT [113]. Chemerin is able to recruit immune cells and release pro-inflammatory cytokines (TNF-α, IL-1β, IL-6) [5,55,57,113]. TNF-α inhibits insulin signal transduction [114]. Studies on *Psammomys obesus* have shown that IL-1β-induced apoptosis of pancreatic islet cells reduces insulin production, potentially exaggerating the level of insulin resistance through subsequent hyperglycemia [106]. Similarly, IL-6 contributes to β-cell loss and impaired insulin signaling [115]. Chemerin-attracted immune cells secrete inflammatory mediators that contribute to diminished insulin signaling, highlighting chemerin’s indirect role in modulating insulin sensitivity. Additionally, chemerin may activate enzymes such as ERK1/2, p38 MAPK, and NF-κB, further promoting insulin resistance. Notably, ERK1/2 inhibition is crucial for maintaining insulin sensitivity, suggesting that chemerin-induced activation of this pathway reduces insulin responsiveness [89].

Individuals with MeS exhibit an altered lipoprotein profile [116]. Dyslipidemia in MeS is characterized by increased low-density lipoprotein (LDL) and decreased high-density lipoprotein HDL levels [117]. Chemerin concentrations positively correlate with small, dense LDL and inversely with mean HDL size. Additionally, chemerin and apolipoprotein B (ApoB) levels increase concurrently, whereas apolipoprotein A-I (ApoA-I) decreases [98]. Notably, subjects diagnosed with MeS are characterized by a higher chemerin-to-HDL ratio [110]. Chemerin-mediated insulin resistance promotes lipolysis in visceral fat, increasing free fatty acid uptake by the liver. This in turn enhances hepatic triglyceride (TG) synthesis, leading to elevated TG levels [33,110,118]. Impaired insulin sensitivity reduces insulin’s ability to suppress ApoB synthesis, resulting in the formation of very low-density lipoproteins (VLDL) [119]. The subsequent lipolysis of TG-rich VLDL generates small, dense, atherogenic LDL particles [120].

Inflammation associated with obesity can alter the lipid profile of HDL by modifying cholesterol esters and cholesterol levels, promoting the synthesis of small and intermediate HDL particles. Furthermore, serum chemerin inversely correlates with the ApoA-I/HDL antioxidant capacity [98]. Given the oxidative stress hypothesis of atherosclerosis and HDL’s antioxidant role, research has focused on paraoxonase 1 (PON1), a key enzyme responsible for HDL’s antioxidant properties [121,122,123]. PON1 inhibits the oxidation of HDL and LDL, offering protection against atherosclerosis, a common MeS-related complication [124,125]. Studies indicate a positive correlation between chemerin and oxidized LDL levels. In obese patients, PON1 activity is negatively associated with circulating chemerin. Since both are expressed in the liver, it is hypothesized that chemerin may influence PON1 production, contributing to MeS-related diseases [60]. Adiponectin, an anti-inflammatory and anti-atherosclerotic protein, is associated with a lower risk of CVDs and MeS [126,127]. The study conducted by Chu SH et al. on 92 patients with an average BMI around 28 kg/m^2^ has demonstrated that high chemerin and low adiponectin levels may contribute to dyslipiedmia and MeS development (OR: 5.79, 95% CI: 1.00–33.70) as compared to the other three groups (high chemerin/high adiponectin; low chemerin/high adiponectin and low chemerin/low adiponectin) [128].

Physical activity and diet can improve MeS-related conditions [129,130]. A study on children found significantly higher chemerin levels in obese subjects compared to non-overweight controls. Lifestyle interventions effectively reduced both chemerin levels and MeS risk factors [131]. Additionally, muscle strength and regular sports activity play a crucial role in MeS prevention, as improvements in muscular strength reduce all components of MeS [132].

## 7. The Role of Chemerin in the Pathophysiology of Hypertension

Chemerin exerts a wide range of effects on the cardiovascular system; it especially contributes to hypertensive reaction development. One of its primary actions is its modulation of the immune system. Functioning as a chemoattractant, chemerin guides macrophages and immature DCs to sites of tissue injury—a process mediated by CMKLR1 [5]. Notably, studies have demonstrated that the chemoattractant activity of chemerin, alongside CMKLR1 expression, allows for the differentiation of plasmacytoid DCs from their myeloid counterparts in human blood [133].

Inflammation is a well-established contributor to hypertension, and chemerin has been shown to correlate strongly with several levels of inflammatory markers, including hs-CRP, TNF-α, IL-6, resistin, and leptin [59]. In addition, serum chemerin levels demonstrate a significant positive correlation with blood pressure, positioning it as a contributing factor in the pathogenesis of hypertension [117,134,135,136,137].

Chemerin exhibits vasoconstrictive properties, contributing to elevated blood pressure through both direct and synergistic mechanisms. It not only causes vasoconstriction independently but also amplifies the actions of other vasoconstrictors [138,139,140,141,142]. Chemerin is a major contributor to vascular remodeling due to vascular smooth muscle cell (VSMC) promotion and proliferation [141,143,144]. Collectively, these mechanisms highlight chemerin’s multifactorial involvement in blood pressure regulation and hypertension pathogenesis (Figure 2).

## 8. Chemerin and Hypertension—Studies

### 8.1. Chemerin in Experimental Models of Hypertension

A substantial body of evidence from animal studies supports the role of chemerin in the pathophysiology of hypertension [134,138,140,144,145,146]. Early investigations demonstrated that chemerin expression is upregulated in glomeruli in the experimental models of hypertensive nephropathy and glomerulonephritis. In the study conducted by Mocker A et al., the role of chemerin in the pathophysiology of hypertension in experimental models was analyzed. The study has investigated the role of chemerin in two rat models of kidney injury–hypertensive nephropathy [2-kidney-1-clip (2k1c)] and in the model of glomerulonephritis (Thy1.1 nephritis) to understand local vs. systemic expression of chemerin and its correlation with inflammation and fibrosis [147]. Both models were characterized by significantly increased chemerin expression in the kidney. In the 2k1c model, chemerin was localized to tubulo-interstitium; however, in the hypertensive nephropathy model, chemerin was localized in the glomeruli. Surprisingly, plasma chemerin levels remained unchanged, indicating a lack of systemic response in early/moderate disease. Additionally, significant correlations were found between renal chemerin expression and parameters of kidney function such as serum creatinine and urea, markers of inflammation (M1 macrophages and neutrophils), and fibrosis (TGF-β1, fibronectin, and collagens I/III/IV) [147]. This upregulation is associated with increased inflammation and fibrosis, both of which are key contributors to renal damage and subsequent hypertension [147].

Chemerin is also able to modulate central blood pressure. Chemerin further promotes VSMC proliferation and migration via ROS-dependent activation of Akt and ERK pathways. These processes are mediated through CMKLR1 and have also been observed in human aortic smooth muscle cells [141]. Prolonged intraperitoneal administration of chemerin for six weeks significantly increased systolic blood pressure (SBP) in murine models, confirming its hypertensive effects. Additionally, chemerin induces VSMC proliferation and migration in a dose- and time-dependent manner via activation of autophagy-related pathways. Elevated levels of autophagy markers (LC3 and beclin-1), as well as increased autophagosome formation, were observed. In metabolically hypertensive rats, chemerin and CMKLR1 expression were significantly upregulated in the medial layers of the thoracic aorta and mesenteric arteries, suggesting a direct role of the chemerin/chemerin 1 axis in vascular remodeling and hypertension development [143].

Chemerin’s vasoconstrictive properties have been substantiated in both human and rat models [138,139]. The adipokine potentiates the vasoconstrictive responses elicited by agents such as endothelin, phenylephrine, and norepinephrine [141,142]. Notably, chemerin-induced vasoconstriction is enhanced following endothelial denudation, implicating a modulatory role of the endothelium [140]. These effects are primarily mediated by CMKLR1, with little involvement from chemerin 2. Both receptors are present in the VSMC and in small resistance vessels and large arteries within the human circulatory system. In rats, chemerin similarly increases systemic blood pressure via CMKLR1 activation. Conversely, pharmacological inhibition of this receptor using CCX832 abolishes the vasoconstrictive response [140]. Perivascular adipose tissue (PVAT), localized around blood vessel walls, regulates vascular homeostasis [148]. Chemerin has been identified in PVAT, suggesting its involvement in vascular function, including blood pressure regulation [149]. Postganglionic sympathetic nerve fibers are located at the junction between myocytes and adipocytes in the outer tunica media [150]. Acting through CMKLR1, chemerin modulates blood vessel contraction via sympathetic activation, leading to vasoconstriction [139]. Furthermore, chemerin is essential for controlling blood flow and systemic vascular resistance [151].

Beyond vascular tone regulation, chemerin contributes to vascular remodeling. Endothelial nitric oxide synthase (NOS3), critical for maintaining vascular homeostasis, is expressed not only in the vessel wall but also in PVAT [152,153]. In mice with adipocyte-specific *NOS3* knockout fed with a high-fat diet, serum chemerin levels and PVAT chemerin expression were significantly elevated. These changes were associated with increased markers of vascular remodeling, indicating that NOS3 plays a regulatory role in adipose tissue-derived chemerin and vascular structure maintenance [144].

Sex-specific differences in chemerin expression and blood pressure regulation have also been reported. In a chemerin knockout rat model, both male and female knockout rats were subjected to a hypertensive challenge using deoxycorticosterone acetate (DOCA) and seawater. Male knockout rats exhibited higher blood pressure compared to wild-type males, whereas female knockout rats demonstrated lower blood pressure than wild-type females. These findings suggest a sex-dependent role of chemerin in modulating blood pressure responses [154].

In one study, rats were allocated to either a high-fat or a high-salt diet. Administration of antisense oligonucleotides (ASOs) targeting chemerin mRNA revealed diet-specific effects on blood pressure. Systemic ASO administration significantly lowered blood pressure in high-fat diet-fed rats, whereas a less pronounced reduction was observed in rats on a high-salt diet. Liver-specific ASO administration yielded minimal changes, indicating that adipose tissue-derived chemerin, rather than hepatic chemerin, plays a predominant role in diet-induced hypertension [145].

### 8.2. The Role of Chemerin in Human Studies

Clinical studies corroborate the experimental findings on chemerin’s involvement in hypertension. Circulating chemerin levels are significantly elevated in patients with hypertension. It was observed that individuals with T2DM and hypertension have presented higher plasma chemerin levels compared to patients with type 2 diabetes alone and healthy controls. Although the sample size was small (31 hypertensive patients with newly diagnosed T2D vs. 81 patients with newly diagnosed T2D and normal blood pressure), it was shown that the serum chemerin level elevation was independent of other variables, including diastolic blood pressure, waist circumference, post-glucose-load insulin levels, and HbA1c [155].

Moreover, chemerin levels in newly diagnosed hypertensive patients were significantly higher than in normotensive controls and positively correlated with inflammatory markers (hs-CRP, TNF-α, and IL-6), obesity parameters, plasma TGs, and HOMA-IR index. Interestingly, although chemerin levels seemed to be independently associated with hypertension when metabolic variables were controlled, this association was lost upon adjustment for inflammatory markers, indicating that inflammation mediates the chemerin–hypertension relationship [117]. The study was conducted on a group of 237 patients with newly diagnosed primary hypertension, defined as first diagnosed SBP higher than 140 mmHg or diastolic blood pressure higher than 90 mmHg.

Chemerin appears to be implicated in angiogenesis—a pathological feature of vascular injury in hypertension. In a family-based study involving 1354 randomly selected patients without regard to phenotype or the disease status of individuals of Mexican–American descent, it was found that plasma chemerin levels were highly heritable [41]. In a population-based cohort of 4101 healthy adults without hypertension, diabetes, dyslipidemia, or renal impairment, it was observed that chemerin levels were positively correlated with both systolic and diastolic blood pressure. Chemerin was also associated with anthropometric indices, HbA1c, total cholesterol, triglycerides, LDL, low HDL, and markers of renal dysfunction [117]. Furthermore, chemerin levels were significantly higher in hypertensive non-dipper patients (i.e., those with <10% nocturnal blood pressure decline) compared to dippers and normotensive individuals. The study group included 60 hypertensive patients compared to 30 healthy controls. Non-dippers exhibited more frequent endothelial dysfunction and target organ damage, suggesting that elevated chemerin may contribute to these adverse outcomes [136]. Moreover, in children with obesity but without clinically diagnosed hypertension, serum chemerin levels were positively correlated with SBP, indicating a possible predictive role in early cardiovascular risk [137].

In pregnant women during the third trimester (around 35 weeks), Chen Y et al. observed significantly elevated chemerin levels in patients with eclampsia relative to those with uncomplicated pregnancies. Elevated chemerin levels were also positively correlated with the development of postpartum hypertension, defined as blood pressure either ≥130/80 mmHg or ≥140/90 mmHg, particularly in patients with preeclampsia [156].

In conclusion, chemerin exerts multifaceted effects on the cardiovascular system and contributes to the pathogenesis of hypertension via mechanisms including inflammation, vasoconstriction, vascular remodeling, and angiogenesis. Evidence from both animal and human studies underscores its potential as a diagnostic and therapeutic target in hypertension.

To summarize, animal studies clearly delineate chemerin’s role in BP elevation through specific molecular and cellular mechanisms, including VSMC remodeling, ROS generation, and CMKLR1 signaling. In contrast, human studies primarily establish correlations between circulating chemerin levels and hypertension-related outcomes, often in the context of inflammation and metabolic dysfunction.

In experimental models of hypertension, chemerin frequently acts locally—such as in the kidney or PVAT—without affecting systemic plasma levels. Human research, however, focuses mainly on circulating chemerin as a potential biomarker.

Whereas animal studies show chemerin directly contributes to vascular remodeling and BP elevation, human data suggest a more indirect, inflammation-mediated role. Preclinical models also provide proof-of-concept for therapeutic interventions—such as CMKLR1 antagonists or antisense therapies—while human studies have yet to test these strategies clinically.

Overall, chemerin emerges as a multifaceted regulator of hypertension. Its causal role is well supported in animal models, while in humans, it serves as a promising biomarker linked to early hypertension and systemic inflammation. Bridging this translational gap will require well-designed interventional studies to determine whether targeting chemerin pathways can effectively prevent or treat hypertension in human populations.

## 9. The Role of Chemerin in the Pathophysiology of Atherosclerosis

Despite significant advancements in CVD management over the past several decades, acute myocardial infarction (AMI) is a leading cause of CVD mortality all over the world. This persistent burden underscores the ongoing need for novel therapeutic approaches and molecular targets. Among emerging candidates, chemerin—a multifunctional adipokine—has garnered considerable attention due to its diverse roles in the pathophysiology of atherosclerosis. The aim of this paragraph was to synthesize current knowledge regarding chemerin at the molecular level, elucidate its influence on the initiation and progression of atherosclerosis, and evaluate its potential applications in both the diagnostic process and therapeutic strategies for atherosclerotic cardiovascular diseases (ASCVD).

Chemerin exhibits pleiotropic biological activity through interactions with the immune system, ECs, adipocytes, and various other tissues. Functioning primarily as a chemoattractant for immune cells, chemerin exerts its effects through binding to its receptors, ChemR23 and CCRL2. These interactions activate critical intracellular signaling pathways—including MAPK, NF-κB, and Janus kinase/signal transducer and activator of transcription (JAK/STAT) pathways—leading to the upregulation of adhesion molecules such as vascular cell adhesion molecule-1 (VCAM-1), intercellular adhesion molecule-1 (ICAM-1), and E-selectin within ECs. This in turn promotes leukocyte adhesion and transmigration, contributing to local inflammation and endothelial injury. Additionally, chemerin facilitates the release of pro-inflammatory cytokines, particularly TNF-α and IL-1β, further amplifying the inflammatory cascade.

### The Role of Chemerin in the Pathophysiology of Atherosclerosis in the Experimental Models

Recent studies have elucidated the role of chemerin in vascular pathology. The brilliant study conducted by Tang C. et al. has shown that chemerin, commonly linked to inflammation and metabolic function, is recruited by CCRL2 to vascular regions prone to atherosclerosis. The CCRL2 is an atypical chemoattractant receptor found on endothelial cells [157]. In the pathological condition of disturbed blood flow (d-flow), common at arterial branch points, the CCRL2 is upregulated in endothelial cells acting not as typical chemokine receptors but as binding and concentrating chemerin, leading to the rapid increase of its local availability [157]. In addition, chemerin exhibits protein disulfide isomerase (PDI)-like enzymatic activity, which can reduce disulfide bonds. As a result, this enzymatic action allows chemerin to activate β2 integrins on monocytes, facilitating their firm adhesion to endothelial cells. Instead of activating its classic receptor CMKLR1, chemerin interacts directly with β2 integrins on monocytes, and it activates ERK1/2 signaling, enhancing monocyte adhesion—a critical early step in plaque development. It can be postulated that this alternative adhesion pathway, independent of G-protein signaling, can be crucial for monocyte recruitment in atherosclerosis [157].

Endothelial progenitor cells (EPCs) play a major role in vascular wall repair and in the pathogenesis of atherosclerosis. The study found that chemerin enhances EPC adhesion, migration, and proliferation with the p38 mitogen-activated protein kinase (MAPK) while reducing apoptosis—potentially aiding initial endothelial repair but contributing to pathological plaque buildup [158]. Additionally, the *ApoE^−/−^* mice model treated with chemerin was characterized by increased lipid accumulation in the aorta, reduced collagen, and increased foam cells, contributing to unstable plaques [158].

Furthermore, chemerin contributes to lipid accumulation and the pathological development of atherosclerotic plaques [158]. Collectively, these mechanisms contribute to the progression of atherosclerosis and its associated cardiovascular complications.

## 10. Chemerin’s Role in Human Atherosclerosis

While preclinical data are compelling, translation into clinical practice remains imperative. Nonetheless, emerging clinical studies offer promising insights into chemerin’s potential utility. In several investigations conducted in Chinese populations—encompassing approximately 500 patients—elevated serum chemerin concentrations were independently associated with coronary artery disease (CAD). Notably, chemerin levels correlated with angiographic evidence of significant coronary stenosis, defined as ≥50% luminal narrowing in at least one major coronary artery in symptomatic patients [159,160,161].

Higher chemerin levels were correlated with coronary artery disease burden [160,162]. A strong correlation was also established with coronary artery calcium score [161]. Although chemerin was not identified as an independent risk factor for multivessel disease, these findings suggest its potential value not only as a qualitative biomarker but possibly as a quantitative parameter in the diagnostic and prognostic stratification of CAD [160,161,162]. In addition to its diagnostic implications, chemerin has been proposed as a prognostic biomarker in both CAD and chronic heart failure [159,163]. In a prospective cohort study of over 800 patients, Zhou et al. demonstrated that elevated serum chemerin levels were predictive of major adverse cardiovascular events (MACEs) and independently associated with all-cause mortality in patients with chronic heart failure [163].

Further research has demonstrated a concurrent increase in chemerin and VEGF expression with epicardial fat volume in patients with CAD [164]. Given the established role of epicardial fat in promoting CAD, plaque progression, and MACEs, this finding further supports chemerin’s potential prognostic relevance. Wu et al. also reported that chemerin and VEGF are independent predictors of vascular remodeling, a process associated with changes in lumen diameter and arterial wall structure, which predisposes to atherosclerosis progression [165,166]. On the contrary, other studies have not demonstrated any important associations between serum chemerin levels and coronary atherosclerosis [59]. However, these studies did report strong associations between chemerin and systemic inflammatory markers, as well as features of metabolic syndrome [59]. Motawi et al. confirmed elevated chemerin levels in patients with CAD, especially in subjects suffering from T2DM and obesity [167]. Conversely, another study has demonstrated that in patients with end-stage renal injury, chemerin has anti-calcification properties [168].

In addition, Lachine et al. have found that elevated chemerin levels in patients with T2DM (without overt CAD) were characterized by carotid intima–media thickness (C-IMT) [169]. Similar findings were reported in patients with newly diagnosed essential hypertension, suggesting that circulating chemerin levels may reflect early endothelial dysfunction [170].

Expanding the scope to cerebrovascular disease, Zhao et al. identified serum chemerin as a risk factor for ischemic stroke [171]. This specificity underscores chemerin’s relevance in stroke patients with underlying atherosclerosis.

In general, chemerin may hold promise in personalized molecular medicine. Although still speculative, genetic evidence linking elevated chemerin levels with increased CAD risk has been identified through Mendelian randomization analysis [172]. These findings emphasize the role of genetic predisposition in atherosclerotic disease and may pave the way for future gene-targeted therapeutic strategies.

In conclusion, while further clinical validation is required, the accumulated evidence strongly supports chemerin as a biomarker with diagnostic, prognostic, and therapeutic potential in atherosclerotic cardiovascular disease. Its incorporation into clinical practice could enhance risk stratification and facilitate targeted molecular therapies aimed at mitigating the global burden of atherosclerosis.

## 11. Clinical Approaches and Future Directions

### 11.1. Diagnostic Approach of Chemerin

Chemerin is a fat-derived signaling protein (adipokine) that has been increasingly studied for its association with hypertension, obesity, diabetes, cardiovascular disease, and kidney dysfunction. Numerous human studies have shown that circulating levels of chemerin are elevated in these conditions, suggesting it could serve as a biomarker. Specifically, chemerin levels tend to increase with disease severity and correlate strongly with visceral fat rather than subcutaneous fat.

Despite this, the use of chemerin as a diagnostic tool faces several limitations. It exists in multiple biologically active and inactive isoforms, but standard clinical assays do not distinguish between them. Furthermore, research indicates that chemerin’s effects on blood pressure and vascular health are often local (e.g., from perivascular fat) rather than systemic, calling into question the relevance of blood levels as a direct measure of disease activity. Sex differences and ethnic variability in baseline levels also complicate its interpretation.

Overall, while chemerin shows promise as a supportive biomarker for cardiometabolic health, its current clinical utility is limited by technological and biological complexities. More precise assays and a deeper understanding of tissue-specific roles are needed before they can be widely used in diagnostics.

### 11.2. Therapeutic Approach

To summarize, it has become evident that chemerin is a major player in inflammation and vascular remodeling. Taking into account the data from the literature, chemerin represents a promising biomarker for both the early detection and prognosis of CVDs, especially atherosclerosis and hypertension.

Starting from the level of basic science, this work presents the current knowledge on the molecular action of chemerin, taking into account its influence on the immune system and inflammatory response. Then, referring to a number of studies on animal models (Table 1), it sheds light on the role of chemerin in the pathogenesis of obesity, hypertension, and atherosclerosis. Finally, the collected observations are reflected in the presented clinical studies (Table 2), which allows for further considerations on potential points of engagement for new therapies in the treatment of cardiovascular diseases. This approach allows for a broader picture to be created, taking into account the available knowledge on chemerin at multiple levels of scientific research.

Importantly, patients with ACS are characterized by significantly higher plasma chemerin concentrations in comparison with subjects diagnosed with stable angina pectoris [179]. Furthermore, there is a correlation between larger infarct size and poorer prognoses, as opposed to cases of unstable angina [179]. Moreover, higher chemerin concentrations are associated with higher risks of complications such as recurrent myocardial infarction, heart failure, and cardiac remodeling [180]. It can be postulated that chemerin might play a dual role—both as a marker and as a mediator—in the progression of AMI and its long-term sequelae. Data from randomized, double-blind, placebo-controlled clinical trials such as CANTOS (Canakinumab Anti-inflammatory Therapy for Atherosclerosis and Cancer Evaluation) have shown that targeting inflammation—independent of lipid levels—could reduce cardiovascular risk [181]. It is suggested that residual inflammatory risk, not just cholesterol, plays a role in the pathophysiology of CVDs. Currently, chemerin is increasingly being explored as a therapeutic target in CVDs. Despite the fact that potential treatment options are still in the preclinical phase, there is mounting pressure to gain more knowledge evaluating the role of inflammation, endothelial dysfunction, and lipid regulation in CVDs. It can be postulated that the chemerin/ChemR23 signaling pathway is of great importance and takes part in plaque stability and vascular remodeling. This pathway exaggerates the progression of atherosclerosis by influencing macrophages and VSMCs. Also, inhibiting chemerin’s receptor, ChemR23, with a small molecule, CCX832, might diminish the downstream inflammatory effect, reduce oxidative stress, and worsen insulin signaling in diabetes, suggesting it could be targeted to improve cardiovascular health in metabolic diseases.

Notably, ASOs-synthetic nucleic acid sequences engineered to bind to specific chemerin mRNA, leading to its degradation and blocking its translation, can be analyzed as a potential therapeutic strategy for controlling hypertension, particularly in obesity-related cases. By integrating chemerin measurements into routine clinical practice, physicians may enhance diagnostic precision, improve patient stratification, and tailor therapeutic interventions to reduce long-term cardiovascular risk. However, more studies are needed to confirm these findings.

## 12. Strengths and Limitations

This study provides a multifaceted structured approach to the issue of the position of chemerin in the pathophysiology of cardiovascular diseases. Starting from the level of basic science, this work presents the current knowledge on the molecular action of chemerin, taking into account its influence on the immune system and inflammatory response. Then, referring to a number of studies on animal models, it sheds light on the role of chemerin in the pathogenesis of obesity, hypertension, and atherosclerosis. Finally, the collected observations are reflected in the presented clinical studies, which allows for further considerations on potential points of engagement for new therapies in the treatment of cardiovascular diseases. This approach allows for a broader picture to be created, taking into account the available knowledge on chemerin at multiple levels of scientific research.

While every effort was made to ensure the quality of this review, several limitations must be acknowledged. Firstly, while included articles were carefully selected for their valuable contribution to the discussion about chemerin, some articles may have been omitted, as studies with significant findings are more likely to be published in reputable journals than those with equivocal results, which may distort the available evidence. Furthermore, applied methodologies often vary significantly, making direct comparisons challenging. Although most studies reach consistent conclusions, due to differences in sample sizes and characteristics of populations, the results should be interpreted attentively. The aforementioned heterogeneity can restrict the applicability of findings to broader contexts. Lastly, due to ongoing research on chemerin, some studies may eventually become outdated over time. These limitations underscore the need for cautious interpretation of the literature and highlight areas for future research to strengthen the evidence base.

## 13. Conclusions/Summary

This review summarizes the role of adipokine chemerin as a critical link between obesity, inflammation, and cardiovascular diseases. Through its pleiotropic effects on immune cells, adipocytes, and vascular cells, chemerin plays an active role in the pathophysiology of metabolic syndrome, hypertension, and atherosclerosis. Chemerin contributes significantly to cardiovascular pathology through several mechanisms. It promotes systemic inflammation and insulin resistance while also causing vasoconstriction and amplifying the effects of other vasoconstrictors. Additionally, it drives vascular remodeling by stimulating the growth of VSMCs. By triggering the adhesion of monocytes to the endothelium, it initiates a crucial step in the development of atherosclerotic plaques. As a central regulator of glucose and lipid balance, targeting chemerin function offers a promising therapeutic strategy for managing inflammation, insulin resistance, and comorbidities associated with obesity. Moreover, its strong associations with cardiovascular conditions suggest chemerin’s potential as a valuable biomarker.

## Figures and Tables

**Figure 1 biomedicines-13-01481-f001:**
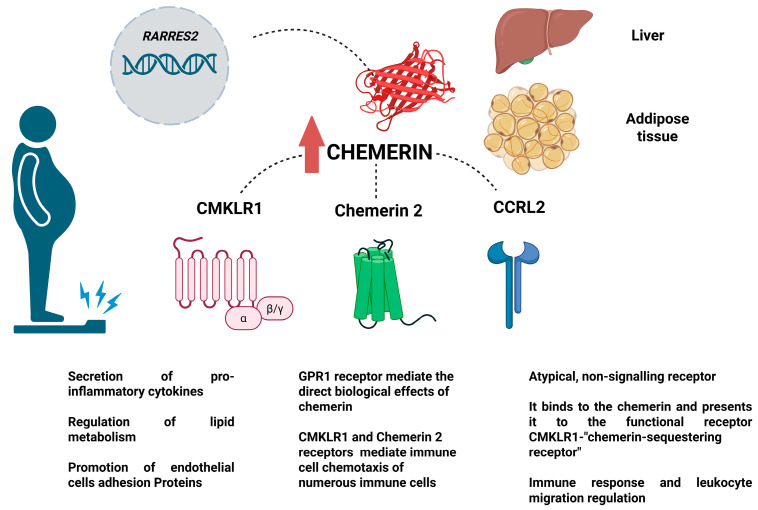
Figure showing the most important chemerin mechanism of action—the main source of chemerin synthesis is adipose tissue and liver. Chemerin, also known as tazarotene-induced gene (*TIG2*; *RARRES2*), acts via three kinds of receptors: CMKLR1 (known also as chemerin 1 or ChemR23) and chemerin 2 (G-protein-coupled receptor 1-GPR1), mediating the direct biological effect of chemerin. In addition, there is a CCRL2, which is described as a chaperone for chemerin. The figure was created in BioRender. Imiela, A.M.

**Figure 2 biomedicines-13-01481-f002:**
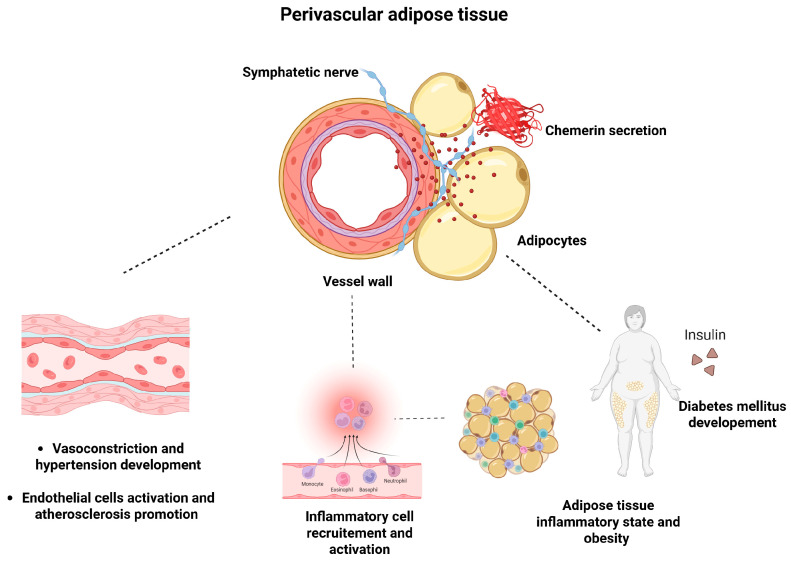
Figure presents the role of perivascular adipose tissue (PVAT) in the pathogenesis of obesity and cardiovascular complications. PVAT serves as a local source of chemerin, which may activate chemerin 1 receptors on sympathetic nerves or smooth muscle cells to promote vascular contraction. Hypertension might lead to endothelial cell injury and activation of endothelial cells’ adhesive molecules, finally promoting atherosclerosis. Higher local chemerin production results in inflammatory cell recruitment and inflammation promotion in white adipose tissue. Obesity and metabolic syndrome development results in poor blood pressure and glycemic control. All the above-mentioned processes promote pro-inflammatory state and contribute to the vicious circle. The figure was created in BioRender. Imiela, A.M.

**Table 1 biomedicines-13-01481-t001:** Chemerin in animal models.

Study	Species	Year	Conclusions
Obesity			
Goralski et al. [17]	C57/BL/6J, *Lep* (ob/ob) mice; mouse 3T3-L1 preadipocytes	2007	Maturing 3T3-L1 cells increase chemerin/*CMKLR1* mRNA and secrete more bioactive chemerin. White adipocytes source and target chemerin signaling.
Bozaoglu et al. [49]	*Psammomys obesus* rats	2007	Chemerin is synthesized by mature adipocytes but not preadipocytes. In obese and T2D *P. obesus*, *chemerin* and *CMKLR1* expression were elevated in adipose tissue compared to lean, normoglycemic *P. obesus*.
Ernst et al. [33]	*Lep* (ob/ob), *Lepr* (db/db), C57BL/6 mice	2010	Significantly higher serum chemerin levels and elevated mRNA levels of *chemerin* and its receptors in liver, skeletal muscle, WAT were observed in mouse models of obesity/diabetes.
Muruganandan et al. [16]	C57BL/6 mice	2011	*CMKLR1* deletion inhibits adipogenesis during differentiation. *Chemerin*/*CMKLR1* deletion leads to loss of cyclin A2/B2 mRNA and protein, inhibiting adipogenesis at the G2/M phase.
Ernst et al. [29]	*CMKLR1* knockout and wild-type mice	2012	Fasting blood glucose, serum insulin, and food consumption were decreased in CMKLR1^−/−^ mice compared to the control group.
Nakamura et al. [86]	Male C57BL/6 mice (WT), Male Sprague Dawley (SD) rats	2018	Chemerin promotes angiogenesis. Corneal assay—chemerin led to a significant increase in corneal neovascularization. Chemerin significantly accelerated neovascularization in the rat aortic ring assay.
Jiang et al. [92]	Female athymic Balb/c nude mice	2018	Chemerin promoted preadipocyte proliferation and the expression of *VEGF*, *FABP4*, and *CMKLR1*, as well as the phosphorylation of proteins in the Akt/mTOR and ERK1/2 pathways, in a concentration-dependent manner.
Hypertension			
Lobato N.S. et al. [142]	Aortic rings of Wistar rats incubated with chemerin	2012	Chemerin potentiates vasoconstriction induced by other vasoconstrictors such as endothelin and phenylephrine or norepinephrine.
Watts S.W. et al. [138]	Sprague Dawley rats, Wistar Kyoto rats, stroke-prone spontaneously hypertensive rats	2013	Chemerin has been shown to have significant vasoconstrictor effects in humans and rats.
Kunimoto H. et al. [141]	Kurabo human aortic SMCs, BALB/c mice, Wister rats mesenteric artery SMCs	2015	Chemerin/CMKLR1 stimulates SMC proliferation and migration via reactive oxygen species-dependent phosphorylation of Akt/ERK, which may lead to vascular structural remodeling and an increase in systolic blood pressure.
Kunimoto H. et al. [141]	Kurabo human aortic SMCs, BALB/c mice, Wister rats mesenteric artery SMCs	2015	Chemerin potentiates vasoconstriction induced by other vasoconstrictors such as endothelin and phenylephrine or norepinephrine.
Darios E.S. et al. [139]	Sprague Dawley rats	2016	Chemerin has been shown to have significant vasoconstrictor effects in humans and rats.
Kennedy A.J. et al. [140]	Sprague Dawley rats	2016	The potent vasoconstrictor effects of chemerin are mediated by CMKLR1, not GPR1.
Watts S.W. et al. [154]	CRISPR/Cas9-induced chemerin knockout Sprague Dawley rats	2018	Chemerin expression shows sex differences. Chemerin is able to modify blood pressure in response to a hypertensive challenge. Male chemerin knockout rats had higher blood pressure than male wild-type rats, while female knockout rats had lower blood pressure than female wild-type rats.
Wen J. et al. [143]	Wistar rats; high-salt, high-fat diet	2019	Chemerin stimulates SMC proliferation and migration via autophagy, which may lead to vascular structural remodeling in metabolic hypertension.
Mocker A. et al. [147]	2-kidney 1-clip hypertensive rats, Thy1.1 nephritic rats	2019	Renal chemerin expression is associated with processes of inflammation and fibrosis related to renal damage.
Ferland D.J. et al. [145]	Dahl S rats; high fat vs. high salt diet	2019	Chemerin secreted from adipose tissue is an important pathological factor in hypertension associated with high fat, not high salt diet.
Yamamoto A. et al. [134]	Wistar rats; intracerebroventricular injection of chemerin-9	2020	Chemerin, injected intraventricularly, increases blood pressure in circulatory system via CMKLR1 located in brain.
Yamamoto A. et al. [173]	Wistar Kyoto rats, spontaneously hypertensive rats; intracerebroventricular injection of CMKLR1	2021	Increased protein expression of CMKLR1 in paraventricular nucleus is at least partly responsible for systemic hypertension in spontaneously hypertensive rats.
Andy W.C. Man et al. [144]	Adipocyte-specific nitric oxide synthase-knockout mice; high-fat diet	2023	Adipocyte NOS3 may play an important role in regulating chemerin in adipose tissue. Adipocyte NOS3 is essential for maintaining vascular homeostasis, and its dysfunction contributes to obesity-induced vascular remodeling and hypertension.
Atherosclerosis			
Nakamura N. et al. [86]	C57BL/6 mice, Sprague Dawley rats	2018	Chemerin promotes migration and angiogenic activities mainly through CMKLR1.
Jia J. et al. [158]	*ApoE^−/−^* mice	2020	Chemerin enhances the adhesion and migration abilities of endothelial progenitor cells and increases the instability of plaques and abnormal lipid accumulation presumably through the p38 MAPK pathway.
Tang C. et al. [157]	*ApoE^−/−^* mice	2023	Chemerin promotes atherosclerotic plaque formation via a CCRL2 receptor-β2 integrin axis.

Abbreviations: CMKLR1—chemokine-like receptor 1; WAT—white adipose tissue; T2D—type 2 diabetes; VEGF—vascular endothelial growth factor; FABP4—fatty acid-binding protein 4; Akt—protein kinase B; ERK—extracellular signal-regulated kinase; SMC—smooth muscle cell; MAPK—mitogen-activated protein kinase.

**Table 2 biomedicines-13-01481-t002:** Chemerin in human studies.

Author	Country	Year	Study Design	Total Patients	Main Conclusions
Obesity					
Sell H. et al. [32]	France, Germany	2010	prospective	60	Serum chemerin levels are significantly higher in morbidly obese patients compared to non-obese individuals.
Dong B. et al. [105]	China	2011	prospective	164	In patients with MetS, higher serum chemerin levels are associated with higher BMI, systolic blood pressure, serum triglycerides, and hs-CRP levels.
Chakaroun R. et al. [90]	Germany	2012	cross-sectional and interventional	740	Moderate weight loss 6 months after a calorie-restricted diet significantly reduces serum chemerin levels. Individuals with type 2 diabetes have chemerin serum concentrations 15% higher than healthy individuals.
Lorincz H. et al. [98]	Hungary	2014	observational study	88	Serum chemerin levels are inversely correlated with levels of HDL-C.
Fulop P. et al. [60]	Hungary	2014	case-control	88	Serum chemerin levels are positively correlated with oxidative stress and inflammation markers.
Lorincz H. et al. [98]	Hungary	2014	observational study	88	Obese patients have higher serum chemerin levels than healthy controls.
Fulop P. et al. [60]	Hungary	2014	case-control	88	Serum chemerin levels correlated positively with leptin levels and negatively with adiponectin levels.
Lorincz H. et al. [98]	Hungary	2014	observational study	88	The proportion of small dense LDL subfraction is higher in obese patients than in the control group. Serum chemerin levels are inversely correlated with mean LDL size.
Mehanna E.T. et al. [174]	Egypt	2016	cross-sectional	200	MetS patients show a significantly higher frequency of the minor allele of *chemerin* rs17173608 polymorphism.
Zylla S. et al. [64]	Germany	2017	prospective	3986	Chemerin promotes inflammation by contributing to the development of leukocyte populations, regardless of body mass.
Yang M. et al. [88]	China	2019	case-control	100	Serum chemerin levels are higher in the group with waist-to-stature > 0.55 compared to the group with waist-to-stature ≤ 0.55.
Shafer-Eggleton J. et al. [110]	USA	2020	prospective	50	In MetS, the chemerin:HDL-C ratio is significantly elevated and shows a positive correlation with MetS severity.
Batista A.P. et al. [175]	Brazil	2020	cross-sectional	508	High serum chemerin levels are associated with elevated triglyceride levels and insulin resistance.
Afify A.A. et al. [176]	Egypt	2022	case-control	75	Compared to the control group, serum chemerin concentrations are significantly elevated in participants with obesity.
Hypertension					
Yang M. et al. [155]	China	2010	cross-sectional	174	Serum chemerin levels are significantly increased in hypertensive patients.
Bozaoglu K. et al. [41]	US, Australia	2010	cross-sectional family-based genetic epidemiological study	1354	Chemerin acts as a stimulator of angiogenesis and contributes to vascular damage in hypertension.
Gu P. et al. [135]	China	2014	case-control	347	Chemerin in hypertensive patients is associated with inflammatory markers, as well as with key components of MetS: obesity, plasma triglycerides, and HOMA-IR.
Lu B. et al. [177]	China	2015	observational study	393	Serum chemerin levels are independently associated with arterial function index and early atherosclerotic changes. Elevated serum chemerin levels in hypertensive patients are associated with endothelial dysfunction.
Atherosclerosis					
Lehrke M. et al. [59]	Germany	2009	cross-sectional	303	Serum chemerin levels are associated with levels of hs-CRP, interleukin 6, TNF-α, resistin, and leptin. Chemerin does not predict coronary atherosclerosis.
Hah Y.J. et al. [162]	Korea	2011	cross-sectional	131	Serum chemerin levels have a significant correlation with the degree of coronary artery stenosis in patients with CAD. Chemerin was not an independent risk factor for multiple vessel disease.
Yan Q. et al. [160]	China	2012	case-control	430	Higher serum chemerin levels are associated with metabolic parameters, increased risk of CAD per se, and number of diseased vessels.
Xiaotao L. et al. [161]	China	2012	cross-sectional	188	Higher serum chemerin levels are associated with the presence of CAD. Chemerin levels may reflect the extent of coronary atherosclerosis in the sense of calcium score.
Kim H.M. et al. [178]	Korea	2012	cross-sectional	70	Serum chemerin levels were not significantly different between asymptomatic type 2 diabetic patients with CAD and without CAD.
Gu P. et al. [170]	China	2015	cross-sectional	356	Serum chemerin levels were independently associated with the index of arterial function and early atherosclerosis in essentially hypertensive patients.
Zhao D. et al. [171]	China	2015	case-control	140	Serum chemerin levels may be an independent risk factor for acute ischemic stroke and carotid artery plaque instability.
Lachine N.A. et al. [169]	Egypt	2016	cross-sectional	180	Serum chemerin could be considered a marker of subclinical atherosclerosis in patients with type 2 diabetes.
Motawi T.M.K. et al. [167]	Egypt	2018	cross-sectional	90	Serum chemerin levels could statistically significantly differentiate between non-obese, non-diabetic patients with CAD and obese, diabetic patients with CAD.
Carracedo M. et al. [168]	Sweden	2019	case-control	163	Chemerin signaling through CMKLR1 in vascular smooth muscle cells protects against vascular calcification in chronic kidney disease.
Zhou X. et al. [163]	China	2019	prospective cohort study	834	Serum chemerin levels correlate with risk of major adverse cardiac events in patients with chronic heart failure.
Wu Q. et al. [164]	China	2020	case-control	100	Epicardial fat volume, adiponectin, chemerin, and VEGF are independent risk factors for vascular remodeling. The expression of adiponectin, chemerin, and VEGF can reflect epicardial fat volume.
Wang B. et al. [159]	China	2022	cohort study	152	Serum chemerin levels are significantly increased in patients with coronary artery disease compared to controls. High chemerin levels in those patients increase the risk of major adverse cardiac events.

Abbreviations: hs-CRP—high-sensitivity C-reactive protein; MetS—metabolic syndrome; HDL-C—high-density lipoprotein cholesterol; LDL—low-density lipoprotein; HOMA-IR—homeostasis model assessment-estimated insulin resistance; TNF-α—tumor necrosis factor α; CAD—coronary artery disease; CMKLR1—chemokine-like receptor 1; VEGF—vascular endothelial growth factor.

## Data Availability

No new data were created or analyzed in this study. Data sharing is not applicable to this article.

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
