# Peer review of "Chemerin as a Driver of Cardiovascular Diseases: New Perspectives and Future Directions"

_biomedicines, 2025, doi:10.3390/biomedicines13061481_

Round 1
Reviewer 1 Report
Comments and Suggestions for Authors
This is a well-organized review examining the emerging role of chemerin in cardiovascular disease, covering its mechanisms, involvement in metabolic syndrome, hypertension, atherosclerosis, and therapeutic perspectives. It is structured and flows logically. However, I have some concerns.
- Please add a paragraph in the introduction or a separate "Methods" section describing how the literature was selected, e.g., databases searched, keywords used, date range, and article types included.
- Consider synthesizing findings more critically, highlighting consistencies and discrepancies across studies, particularly when animal and human data diverge.
- Please ensure that all figures are clearly introduced in the main text with appropriate explanation and relevance to the surrounding discussion. For example, “Upon binding to CMKLR1, chemerin activates the Gi protein, reducing cyclic adenosine monophosphate (cAMP) levels and inducing phosphorylation of nuclear factor kappa B (NFκB) and extracellular signal-regulated kinase 1/2 (ERK1/2)” was not illustrated in Figure 1.
- In 8.2, the manuscript lists human studies but does not include details, e.g., sample size, design, and confounding factors. Please add a brief critique of each cited study's design and limitations to provide a balanced interpretation.
- Language needs to be improved for clarity and accuracy.
Author Response
#Review 1
This is a well-organized review examining the emerging role of chemerin in cardiovascular disease, covering its mechanisms, involvement in metabolic syndrome, hypertension, atherosclerosis, and therapeutic perspectives. It is structured and flows logically. However, I have some concerns.
Query 1: Please add a paragraph in the introduction or a separate "Methods" section describing how the literature was selected, e.g., databases searched, keywords used, date range, and article types included.
Answer 1: We are very grateful for this suggestion. The paragraph “Methods" has been added as follows:
“We conducted a comprehensive PubMed search for articles published between 2007 and 2025 using the keywords “chemerin” and “cardiovascular disease”. Studies were selected with particular emphasis on recently published research. Relevant articles were thematically categorized into five major areas: inflammation, obesity, metabolic syndrome, hypertension, and atherosclerosis. Within each thematic category, studies were further divided into subgroups based on whether they involved animal models or human subjects. Findings were then compared to identify consistent conclusions and notable discrepancies, with a particular focus on their potential clinical relevance.”
Q2: Consider synthesizing findings more critically, highlighting consistencies and discrepancies across studies, particularly when animal and human data diverge.
Answer 2: We are enormously grateful for this valuable suggestion. Based on your advice, we have added a paragraph summarizing the potential differences between animal and human studies. Additionally, we have included new content in both the “Obesity” and “Hypertension” sections. All changes are pointed out in red.
“Obesity” section:
Generally, in mouse models, chemerin promotes adipocyte differentiation via CMKLR1, with both increasing during adipogenesis through ERK1/2 signaling. Deletion of chemerin or CMKLR1 halts differentiation at the G2/M phase by reducing cyclins A2 and B2. CMKLR1-deficient mice show lower body mass, food intake, and fat accumulation under various diets but develop glucose intolerance due to impaired insulin secretion, highlighting chemerin's metabolic role. Chemerin also enhances angiogenesis in vivo, as shown in Matrigel plug, aortic ring, and corneal assays, by activating PI3K/Akt, ERK1/2, and MAPK pathways—effects lost when CMKLR1 is silenced, confirming its key role.
Human studies link elevated serum chemerin to higher BMI, waist-to-hip ratio, and adiposity. Obese individuals (BMI >30) have significantly higher chemerin than lean individuals, with levels decreasing post-bariatric surgery. In vitro, chemerin stimulates endothelial migration, tube formation, and angiogenic marker expression (VEGF, FABP4) in HUVECs and HMECs via Akt and ERK pathways—effects abolished by CMKLR1 silencing .Chemerin also contributes to obesity-related inflammation by recruiting macrophages, dendritic cells, and NK cells, and correlates with pro-inflammatory markers such as TNF-α, IL-6, and hs-CRP.
“Hypertension” section:
To summarize, animal studies clearly delineate chemerin’s role in BP elevation through specific molecular and cellular mechanisms, including VSMC remodeling, ROS generation, and CMKLR1 signaling. In contrast, human studies primarily establish correlations between circulating chemerin levels and hypertension-related outcomes, often in the context of inflammation and metabolic dysfunction.
In experimental models of hypertension, chemerin frequently acts locally—such as in the kidney or PVAT—without affecting systemic plasma levels. Human research, however, focuses mainly on circulating chemerin as a potential biomarker.
Whereas animal studies show chemerin directly contributes to vascular remodeling and BP elevation, human data suggest a more indirect, inflammation-mediated role. Preclinical models also provide proof-of-concept for therapeutic interventions—such as CMKLR1 antagonists or antisense therapies—while human studies have yet to test these strategies clinically.
Overall, chemerin emerges as a multi-faceted regulator of hypertension. Its causal role is well-supported in animal models, while in humans, it serves as a promising biomarker linked to early hypertension and systemic inflammation. Bridging this translational gap will require well-designed interventional studies to determine whether targeting chemerin pathways can effectively prevent or treat hypertension in human populations.
Q3: Please ensure that all figures are clearly introduced in the main text with appropriate explanation and relevance to the surrounding discussion.
For example, “Upon binding to CMKLR1, chemerin activates the Gi protein, reducing cyclic adenosine monophosphate (cAMP) levels and inducing phosphorylation of nuclear factor kappa B (NFκB) and extracellular signal-regulated kinase 1/2 (ERK1/2)” was not illustrated in Figure 1.
Answer 3: Thank you for this suggestion. We have removed “Figure 1” from the detailed description of chemerin’s molecular mechanisms. Instead, we have added the following sentence: “Figure 1 summarizes the most relevant mechanisms of chemerin action; its primary sites of synthesis are adipose tissue and the liver.”
Q4: In 8.2, the manuscript lists human studies but does not include details, e.g., sample size, design, and confounding factors. Please add a brief critique of each cited study's design and limitations to provide a balanced interpretation.
Answer 4: Thank you for the suggestion. Paragraph 8.2 has been expanded to provide more detailed information on the cited articles, including key findings and their relevance to the review topic. We have also prepared tables to present the data more comprehensive. All changes are marked in red.
Q5: Language needs to be improved for clarity and accuracy.
Answer 5: We sincerely appreciate this valuable suggestion. We have carefully revised the text, improving the language and making necessary grammatical and stylistic corrections throughout the manuscript.
Reviewer 2 Report
Comments and Suggestions for Authors
Comment 1:
a. Figure 1 need more explanation. For example, "It binds specific chemerin isoforms, including chemerin-9, chemerin-13, and chemerin-21-57, though its precise function remains unclear (Figure 1)" However, there are no explanation about isoforms or chemerin 9,13,21-57 mechanism of action while reading text and figure.
b. Figure 1 needs to add labelling of liver and adipose tissue. Additionally, it is unclear how to read bullet points along with figure 1 and what it explains (e.g. sequence or just key points). The better explanation will help readers to understand underlaying mechanism of figure.
Comment 2:
- Line 190, section 5.1 could be corrected. "Chemerin and Obesity - Data Coming from Animals Models"
- Line 191 starting looks unfinished.
- Some reference in section 5.1 are in-vitro cell (for example line 225-229 ...).
- Section 5.2 need correction in title as follows:
- Chemerin and Obesity - Data Coming from Humans
Comment 3:
Throughout review
- a clear distinction of diagnostic application and therapeutic application would be helpful to readers.
- Tabular presentation of animal data and human data with intended use must be presented creatively.
Comment 4:
- Line 555, patient addition of characteristics (chronic heart failure) would be useful.
- Line 565, CAD shall be replaced with specific word atherosclerosis as per cited reference.
- Line 577-78 can be corrected as: Expanding the scope to cerebrovascular disease, Zhao et al. [173] identified serum chemerin as an risk factor for ischemic stroke [173].
- Few references e.g. 60, 65, 104, 109, 117, 121-125, 168 needs proper formatting.
Author Response
#Review 2
Query 1:
Figure 1 need more explanation. For example, "It binds specific chemerin isoforms, including chemerin-9, chemerin-13, and chemerin-21-57, though its precise function remains unclear (Figure 1)" However, there are no explanation about isoforms or chemerin 9,13,21-57 mechanism of action while reading text and figure.
Answer 1: We appreciate this suggestion. We have corrected the figure as indicated below. We have removed the fragment about isoforms.
- Figure 1 needs to add labelling of liver and adipose tissue. Additionally, it is unclear how to read bullet points along with figure 1 and what it explains (e.g. sequence or just key points). The better explanation will help readers to understand underlaying mechanism of figure.
Answer 1: We would like to thank you for this suggestions. We have labelled the liver and adipose tissue. We have removed the bullet points.
Query 2: Line 190, section 5.1 could be corrected. "Chemerin and Obesity - Data Coming from Animals Models"
- Line 191 starting looks unfinished.
- Some reference in section 5.1 are in-vitro cell (for example line 225-229 ...).
- Section 5.2 need correction in title as follows:
- Chemerin and Obesity - Data Coming from Humans
Answer 2: Thank you for this valuable suggestion. The mentioned corrections were made according to the provided instructions. All changes are marked in red.
Query 3:Throughout review
- a clear distinction of diagnostic application and therapeutic application would be helpful to readers. – A.I.
Answer 3: We have modifed a paragraph describing the diagnostic and therapeutic actions of chemerin. We have also added an entirely new paragraph on the potential diagnostic role of chemerin:
Chemerin is a fat-derived signaling protein (adipokine) that has been increasingly studied for its association with hypertension, obesity, diabetes, cardiovascular disease, and kidney dysfunction. Numerous human studies have shown that circulating levels of chemerin are elevated in these conditions, suggesting it could serve as a biomarker. Specifically, chemerin levels tend to increase with disease severity and correlate strongly with visceral fat rather than subcutaneous fat.
Despite this, the use of chemerin as a diagnostic tool faces several limitations. It exists in multiple biologically active and inactive isoforms, but standard clinical assays do not distinguish between them. Furthermore, research indicates that chemerin's effects on blood pressure and vascular health are often local (e.g., from perivascular fat) rather than systemic, calling into question the relevance of blood levels as a direct measure of disease activity. Sex differences and ethnic variability in baseline levels also complicate its interpretation.
Overall, while chemerin shows promise as a supportive biomarker for cardiometabolic health, its current clinical utility is limited by technological and biological complexities. More precise assays and deeper understanding of tissue-specific roles are needed before it can be widely used in diagnostics.
We have also added a paragraph on therapeutic approach:
To summarize, it has become evident that chemerin is a major player in the inflammation and vascular remodeling. Taking into account the data from the literature, chemerin represents a promising biomarker for both the early detection and prognosis of CVDs, especially atherosclerosis and hypertension.
Starting from the level of basic science, this work presents the current knowledge on the molecular action of chemerin, taking into account its influence on the immune system and inflammatory response. Then, referring to a number of studies on animal models, it sheds light on the role of chemerin in the pathogenesis of obesity, hypertension and atherosclerosis. Finally, the collected observations are reflected in the presented clinical studies, which allows for further considerations on potential points of engagement for new therapies in the treatment of cardiovascular diseases. This approach allows for a broader picture to be created, taking into account the available knowledge on chemerin at multiple levels of scientific research.
Importantly, patients with ACS are characterized by significantly higher plasma chemerin concentration in comparison with subjects diagnosed with stable angina pectoris [179]. Furthermore, there is a correlation between larger infarct size and poorer prognoses, as opposed to cases of unstable angina [179]. Moreover, higher chemerin concentrations are associated with higher risks of complications such recurrent myocardial infarction, heart failure and cardiac remodeling [180]. It can be postulated that chemerin might play a dual role—both as a marker and as a mediator—in the progression of AMI and its long-term sequelae. Data from randomized, double-blind, placebo-controlled clinical trials such as-CANTOS (Canakinumab Anti-inflammatory Therapy for Atherosclerosis and Cancer Evaluation) have shown that targeting inflammation—independent of lipid levels—could reduce cardiovascular risk [181]. It is suggested that residual inflammatory risk, not just cholesterol, plays a role in pathophysiology of CVDs. Currently, chemerin is increasingly being explored as a therapeutic target in CVDs. Despite the fact that potential treatment options are still in the preclinical phase, there is a mounting pressure to gain more knowledge evaluating the role of inflammation, endothelial dysfunction, and lipid regulation in CVDs. It can be postulated that chemerin/ChemR23 signaling pathway is of great importance and takes part in plaque stability and vascular remodeling. This pathway exaggerates the progression of atherosclerosis by influencing macrophages and VSMCs. Also, inhibiting chemerin’s receptor-ChemR23 with a small molecule-CCX832 might diminish downstream inflammatory effect, reduce oxidative stress and worsen insulin signaling in diabetes, suggesting it could be targeted to improve cardiovascular health in metabolic diseases.
Notably, ASOs-synthetic nucleic acid sequences-engineered to bind to specific chemerin mRNA, leading to its degradation and blocking its translation, can be analyzed as a potential therapeutic strategy for controlling hypertension, particularly in obesity-related cases. By integrating chemerin measurements into routine clinical practice, physicians may enhance diagnostic precision, improve patient stratification, and tailor therapeutic interventions to reduce long-term cardiovascular risk. However, more studies are needed to confirm these findings.
Q3: Tabular presentation of animal data and human data with intended use must be presented creatively.
Answer 3: We are very grateful for this suggestion. We have designed a tabular presentation of the results to facilitate comparison of the collected data. New two tables have been added to the main text.
Query 4:
- Line 555, patient addition of characteristics (chronic heart failure) would be useful.
- Line 565, CAD shall be replaced with specific word atherosclerosis as per cited reference.
- Line 577-78 can be corrected as: Expanding the scope to cerebrovascular disease, Zhao et al. [173] identified serum chemerin as an risk factor for ischemic stroke [173].
Answer 4: We are very grateful for this suggestion. The mentioned corrections were made according to the provided instructions.
Query 5: Few references e.g. 60, 65, 104, 109, 117, 121-125, 168 needs proper formatting.
Answer 5: Thank you for this suggestion. We have used EndNote to format the References. All references have been formatted according to MDPI style.
Reviewer 3 Report
Comments and Suggestions for Authors
The review was conducted to explore chemerin's potential role in the pathogenesis of cardiovascular disease (CVD), focusing on its immunomodulatory functions, impact on vascular inflammation, and endothelial dysfunction. The topic is interesting; however, some modifications are necessary.
- The topic does not appear to be a novel review. Several similar reviews have been published in this journal and elsewhere, such as PMID: 36428537 and PMID: 37447205. You should explain why you decided to provide a similar review again.
- The rationale for the review should be explained using related references and highlight the existing data gaps that necessitate this review.
- The introduction is too short and should be expanded to a minimum length of one page.
- Some paragraphs are excessively long. If a paragraph exceeds 200 to 300 words, it is likely too lengthy for most readers to comprehend easily. Breaking it into smaller sub-paragraphs can significantly enhance readability.
- It is recommended that all instances of "we" or "our" be replaced with phrases such as "current study," "this study," or "present study.
- Please add a section for the methodology and describe how you identified related studies. Additionally, to ensure the methodological rigor of the narrative review, it is recommended to utilize the SANRA (Scale for the Assessment of Narrative Review Articles) tool.
- A section should be allocated to discuss the strengths and limitations of the study.
- Please include a separate section for the conclusion and summarize the key points of the review in a single paragraph.
- There are multiple self-citations.
- The similarity percentage is currently 22%. Please reduce it to below 18% to avoid plagiarism.
Extensive proofreading and paraphrasing are required.
Author Response
#Review 3
The review was conducted to explore chemerin's potential role in the pathogenesis of cardiovascular disease (CVD), focusing on its immunomodulatory functions, impact on vascular inflammation, and endothelial dysfunction. The topic is interesting; however, some modifications are necessary.
Q1: The topic does not appear to be a novel review. Several similar reviews have been published in this journal and elsewhere, such as PMID: 36428537 and PMID: 37447205. You should explain why you decided to provide a similar review again.
Answer 1A: While both manuscripts you mentioned address the general role of chemerin in cardiovascular disease and from a nutritional perspective, our review takes a different approach. We aim to provide a cross-sectional summary of chemerin's role in specific cardiovascular conditions, including obesity, hypertension, and coronary artery disease, with a particular emphasis on immune cells and inflammation. Our review explores the emerging role of chemerin in cardiovascular disease, examining its mechanisms, its involvement in metabolic syndrome, hypertension, and atherosclerosis, and discussing potential therapeutic perspectives.
Q2: The rationale for the review should be explained using related references and highlight the existing data gaps that necessitate this review.
Answer 2:
We are grateful for this suggestion. Chemerin has been widely recognized as a key adipokine in obesity and metabolic disorders. However, its direct immunological role in cardiovascular pathogenesis remains underexplored, leaving a critical gap in our understanding. Most existing studies treat chemerin’s cardiovascular effects as a monolith, without examining its specific roles in hypertension, heart failure, atherosclerosis, or myocardial infarction. This lack of specificity has hindered precise mechanistic and therapeutic insights. While emerging clinical data suggest chemerin's potential as a biomarker for cardiovascular outcomes, the evidence remains fragmented. There is an urgent need to synthesize these findings within a cardiovascular framework to elucidate how early immune recruitment by chemerin shapes atherogenesis, vascular inflammation, and myocardial remodeling.
The explanation outlined above has been incorporated into the introduction of the review, as follows:
Chemerin has been widely recognized as a key adipokine in obesity and metabolic disorders. However, its direct immunological role in cardiovascular pathogenesis remains underexplored, leaving a critical gap in our understanding. Most existing studies treat chemerin’s cardiovascular effects as a monolith, without examining its specific roles in hypertension, atherosclerosis, or myocardial infarction. This lack of specificity has hindered precise mechanistic and therapeutic insights. While emerging clinical data suggest chemerin's potential as a biomarker for cardiovascular outcomes, the evidence remains fragmented. There is an urgent need to synthesize thes findings within a cardiovascular framework to elucidate how early immune recruitment by chemerin shapes atherogenesis, vascular inflammation, and myocardial remodeling.
The aim of this manuscript is to address the potential detrimental role of chemerin in the pathogenesis of cardiovascular disorders. This work concentrates on immunomodulatory effects of chemerin and its implications in the vasculature, pro-inflammatory state development and endothelium dysfunction. It places a central and explicit focus on chemerin's immunomodulatory functions in CVD pathogenesis. Particular attention is given to chemerin's function as a chemotactic agent essential for the initial recruitment of immune cells such as dendritic cells, macrophages, and lymphocytes. According to the latest research findings, this review aims not only to clarify the multifactorial role of chemerin in CVD pathophysiology, but also its potential therapeutic and diagnostic applications. It advances the current understanding of chemerin's involvement in CVDs by presenting updated perspectives and identifying key avenues for future research. Consequently, this work offers an updated and comprehensive synthesis that, through its specialized emphasis on immunomodulation, provides an outlook on chemerin's role across various CVDs.
Q3: The introduction is too short and should be expanded to a minimum length of one page.
Answer 3: Thank you for this valuable suggestion. The introduction has been expanded to include more detailed information on the context of the review and the used methodology.
The introduction has been expanded, as follows:
In the past several years, it has become evident that the immune system plays a pivotal role in the pathogenesis of atherosclerosis, heart failure, venous thromboembolism and even systemic hypertension [1-3]. Obesity and concomitant cardiovascular diseases (CVD) are the most common cause of death worldwide. Adipokines, which are produced in the adipose tissue possess a wide plethora of endocrine and immunomodulatory functions [4]. Discovered at the beginning of the 20th century, chemerin is a chemotactic molecule which is involved in an early recruitment of dendritic cells (DC), endothelial cells (EC), macrophages and lymphocytes [5]. Chemerin possess multiple functions regulating cell migration and vasculature homeostasis [6]. Dysregulation of the adipokines profiles result in the series of pathological changes such as low grade inflammation, insulin resistance, metabolic syndrome and poor blood pressure control.
Chemerin has been widely recognized as a key adipokine in obesity and metabolic disorders. However, its direct immunological role in cardiovascular pathogenesis remains underexplored, leaving a critical gap in our understanding. Most existing studies treat chemerin’s cardiovascular effects as a monolith, without examining its specific roles in hypertension, atherosclerosis, or myocardial infarction. This lack of specificity has hindered precise mechanistic and therapeutic insights. While emerging clinical data suggest chemerin's potential as a biomarker for cardiovascular outcomes, the evidence remains fragmented. There is an urgent need to synthesize thes findings within a cardiovascular framework to elucidate how early immune recruitment by chemerin shapes atherogenesis, vascular inflammation, and myocardial remodeling.
The aim of this manuscript is to address the potential detrimental role of chemerin in the pathogenesis of cardiovascular disorders. This work concentrates on immunomodulatory effects of chemerin and its implications in the vasculature, pro-inflammatory state development and endothelium dysfunction. It places a central and explicit focus on chemerin's immunomodulatory functions in CVD pathogenesis. Particular attention is given to chemerin's function as a chemotactic agent essential for the initial recruitment of immune cells such as dendritic cells, macrophages, and lymphocytes. According to the latest research findings, this review aims not only to clarify the multifactorial role of chemerin in CVD pathophysiology, but also its potential therapeutic and diagnostic applications. It advances the current understanding of chemerin's involvement in CVDs by presenting updated perspectives and identifying key avenues for future research. Consequently, this work offers an updated and comprehensive synthesis that, through its specialized emphasis on immunomodulation, provides an outlook on chemerin's role across various CVDs.
Q3: Some paragraphs are excessively long. If a paragraph exceeds 200 to 300 words, it is likely too lengthy for most readers to comprehend easily. Breaking it into smaller sub-paragraphs can significantly enhance readability.
Answer 3: We are very grateful for this suggestion. Paragraph 3 has been divided into smaller parts according to the described aspects of the molecular interaction of chemerin. All changes are marked in red.
Q4: It is recommended that all instances of "we" or "our" be replaced with phrases such as "current study," "this study," or "present study.
Answer 4: The mentioned corrections were made according to the provided instructions. All changes are marked in red.
Q5: Please add a section for the methodology and describe how you identified related studies. Additionally, to ensure the methodological rigor of the narrative review, it is recommended to utilize the SANRA (Scale for the Assessment of Narrative Review Articles) tool.
Answer 5: We sincerely appreciate this thoughtful and constructive suggestion. In response, we have added a dedicated section outlining the Methods used in the current review. Although we did not apply the SANRA (Scale for the Assessment of Narrative Review Articles) tool in this manuscript, we recognize its significance as a standardized framework for assessing narrative reviews. We will certainly consider its application in future work to further enhance the methodological rigor and transparency of our reviews.
METHODS:
A comprehensive PubMed search was performed to analyze the articles published between 2007 and 2025 using the keywords “chemerin” and “cardiovascular disease”. Studies were selected with particular emphasis on recently published research. Relevant articles were thematically categorized into five major areas: inflammation, obesity, metabolic syndrome, hypertension, and atherosclerosis. Within each thematic category, studies were further divided into subgroups based on whether they involved animal models or human subjects. Findings were then compared to identify consistent conclusions and notable discrepancies, with a particular focus on their potential clinical relevance.
Q6: A section should be allocated to discuss the strengths and limitations of the study.
Answer 6: The “Strengths and Limitations” paragraph has been included to highlight advantages and imperfections of the review, as follows:
This study provides a multifaceted structured approach to the issue of the position of chemerin in the pathophysiology of cardiovascular diseases. Starting from the level of basic science, this work presents the current knowledge on the molecular action of chemerin, taking into account its influence on the immune system and inflammatory response. Then, referring to a number of studies on animal models, it sheds light on the role of chemerin in the pathogenesis of obesity, hypertension and atherosclerosis. Finally, the collected observations are reflected in the presented clinical studies, which allows for further considerations on potential points of engagement for new therapies in the treatment of cardiovascular diseases. This approach allows for a broader picture to be created, taking into account the available knowledge on chemerin at multiple levels of scientific research.
While every effort was made to ensure the quality of this review, several limitations must be acknowledged. Firstly, while included articles were carefully selected for their valuable contribution to discussion about chemerin, some articles may have been omitted, as studies with significant findings are more likely to be published in reputable journals than those with equivocal results, which may distort the available evidence. Furthermore, applied methodologies often vary significantly, making direct comparisons challenging. Although most studies reach consistent conclusions, due to differences in sample sizes and characteristics of populations, the results should be interpreted attentively. Aforementioned heterogeneity can restrict the applicability of findings to broader contexts. Lastly, due to ongoing research on chemerin some studies may eventually become outdated over time. These limitations underscore the need for cautious interpretation of the literature and highlight areas for future research to strengthen the evidence base.
Q7: Please include a separate section for the conclusion and summarize the key points of the review in a single paragraph.
Answer 7: We sincerely appreciate this suggestion. The “Conclusions/Summary” paragraph has been included to emphasize most important findings and identify subjects of future research.
This review summarizes the role of the adipokine chemerin as a critical link between obesity, inflammation, and cardiovascular diseases. Through its pleiotropic effects on immune cells, adipocytes, and vascular cells, chemerin plays an active role in the pathophysiology of metabolic syndrome, hypertension, and atherosclerosis. Chemerin contributes significantly to cardiovascular pathology through several mechanisms. It promotes systemic inflammation and insulin resistance while also causing vasoconstriction and amplifying the effects of other vasoconstrictors. Additionally, it drives vascular remodeling by stimulating the growth of VSMCs. By triggering the adhesion of monocytes to the endothelium, it initiates a crucial step in the development of atherosclerotic plaques. As a central regulator of glucose and lipid balance, targeting chemerin function offers a promising therapeutic strategy for managing inflammation, insulin resistance, and comorbidities associated with obesity. Moreover, its strong associations with cardiovascular conditions suggest chemerin's potential as a valuable biomarker.
Q8: There are multiple self-citations.
Answer 8: Thank you for this suggestion. We have been working in the field of immunity of hypertension and cardiovascular disease, however we have removed the citations which are not directly related to this manuscript:
- Imiela, A.M.; Mikolajczyk, T.P.; Guzik, T.J.; Pruszczyk, P. Acute Pulmonary Embolism and Immunity in Animal Models. Arch Immunol Ther Exp (Warsz) 2024, 72, doi:10.2478/aite-2024-0003.
- Imiela, A.M.; Mikolajczyk, T.P.; Pruszczyk, P. Novel Insight into Inflammatory Pathways in Acute Pulmonary Embolism in Humans. Arch Immunol Ther Exp (Warsz) 2024, 72, doi:10.2478/aite-2024-0021.
Q9: The similarity percentage is currently 22%. Please reduce it to below 18% to avoid plagiarism.
Answer 10: We would like to thank you for this suggestion. We have improved the whole text of the manuscript to avoid the similarities. We hope that our modification will improve the quality of our manuscript.
Round 2
Reviewer 3 Report
Comments and Suggestions for Authors
The authors tried to modify the manuscript. It has the potential to be accepted for publication.
Comments on the Quality of English LanguageExtensive proofreading is required.